# The Potential of Scopolamine as an Antidepressant in Major Depressive Disorder: A Systematic Review of Randomized Controlled Trials

**DOI:** 10.3390/biomedicines11102636

**Published:** 2023-09-26

**Authors:** Paweł Moćko, Katarzyna Śladowska, Paweł Kawalec, Yana Babii, Andrzej Pilc

**Affiliations:** 1Health Policy and Management Department, Institute of Public Health, Faculty of Health Sciences, Jagiellonian University Medical College, Skawińska 8, 31-066 Kraków, Poland; pawel.mocko@uj.edu.pl; 2Department of Nutrition and Drug Research, Institute of Public Health, Faculty of Health Sciences, Jagiellonian University Medical College, Skawińska 8, 31-066 Kraków, Poland; katarzyna.wojcieszek@uj.edu.pl (K.Ś.); pawel.kawalec@uj.edu.pl (P.K.); 3Department of Neurobiology, Institute of Pharmacology, Polish Academy of Sciences, Smętna 12, 31-343 Kraków, Poland; babii@if-pan.krakow.pl

**Keywords:** antidepressant, antimuscarinics, clinical trial, depression, hyoscine, major depressive disorder, MDD, muscarinic antagonists, randomized controlled trial, RCT, systematic review, scopolamine

## Abstract

Major depressive disorder is one of the most severe mental disorders. It strongly impairs daily functioning, and, in extreme cases, it can lead to suicide. Although different treatment options are available for patients with depression, there is an ongoing search for novel therapeutic agents, such as scopolamine (also known as hyoscine), that would offer higher efficacy, a more rapid onset of action, and a more favorable safety profile. The aim of our study was to review the current clinical evidence regarding the use of scopolamine, a promising therapeutic option in the treatment of depression. A systematic literature search was performed using PubMed, Embase, and CENTRAL databases up to 5 June 2023. We included randomized placebo-controlled or head-to-head clinical trials that compared the clinical efficacy and safety of scopolamine in the treatment of major depressive disorder. Two reviewers independently conducted the search and study selection and rated the risk of bias for each study. Four randomized controlled trials were identified in the systematic review. The included studies investigated the use of scopolamine administered as an oral, intramuscular, or intravenous drug, alone or in combination with other antidepressants. The results indicated that scopolamine exerts antidepressant effects of varying intensity. We show that not all studies confirmed a statistically and clinically significant reduction of depressive symptoms vs. placebo. A broader perspective on scopolamine use in antidepressant treatment should be confirmed in subsequent large randomized controlled trials assessing both effectiveness and safety. Therefore, studies directly comparing the effectiveness of scopolamine depending on the route of administration are required.

## 1. Introduction

Major depressive disorder (MDD), also referred to as depression, is a mental disorder characterized by a depressed mood, increased feelings of sadness and hopelessness, loss of interest (anhedonia), low energy, fatigue, low self-esteem, inappropriate feelings of guilt, suicidal thoughts, reduced attention span, agitation or decreased mobility, sleep disturbances, and increased or decreased appetite with changes in body weight [1,2]. According to the Diagnostic and Statistical Manual of Mental Disorders, 5th Edition [3], MDD is diagnosed if a patient has at least five of the above symptoms during the same 2-week period, including a depressed mood or anhedonia.

Depression negatively affects daily functioning in several areas. It reduces mental and physical activity, limits the patient’s ability to learn, leads to deterioration in relationships with other people, causes the patient to lose interest in work, reduces work efficiency, and significantly affects the patient’s ability to lead a satisfactory life [4,5]. The quality of life of adults with depression is known to be lower than that of adults with diabetes, hypertension, and chronic lung disease [5]. Depression sometimes occurs together with anxiety and may be a risk factor for other medical conditions such as cardiovascular or immune diseases [6,7,8,9]. Studies indicated a relationship between depressive symptoms and an increased risk of cardiovascular disease or even early death [7,8,9]. Depression also affects the production of hormones such as adrenaline, corticoids, and catecholamine, as well as cytokine homeostasis, which in turn has an impact on the immune system [9]. Moreover, it may lead to suicidal thoughts, suicidal attempts, and even suicide, resulting in increased mortality in the population of patients with depression [10].

Depression is a widespread disease and a global health problem. According to a report by the World Health Organization, approximately 280 million people are affected by depression worldwide, which corresponds to 3.8% of the global population, including 5% of adults [11,12]. Before 2020, depressive and anxiety disorders were already the leading contributors to the global health-related burden. The burden was further increased by the outbreak of the coronavirus disease 2019 (COVID-19) pandemic [13]. Restrictions related to COVID-19 led to an increase in the global prevalence of anxiety and depression by 25% in the first year of the pandemic [14]. A survey conducted in the United States revealed that the prevalence of depression symptoms was more than 3-fold higher during the COVID-19 pandemic than before the pandemic [15]. Due to its high prevalence, depression also constitutes a significant economic burden by generating high direct costs (e.g., those related to hospital stay, medical examinations, medications) and indirect costs (e.g., those related to absenteeism, unemployment, and social care) [16,17].

Therefore, when patients are diagnosed with depression, it is crucial to implement appropriate treatment as soon as possible. The goal of treatment in patients with depression is to achieve a complete therapeutic response and symptom remission as quickly as possible; to prevent the early recurrence of symptoms; and to ensure that the patient returns to normal psychosocial functioning. The choice of treatment depends on the severity of symptoms and the patient’s individual characteristics. When choosing an antidepressant, the following factors should be considered: the risk of side effects, comorbidities, possible interactions with other drugs taken by the patient, the patient’s preferences, any previous antidepressant therapies, treatment costs, and withdrawal symptoms [1,2,18,19].

Despite the numerous pharmacological and nonpharmacological options available for the treatment of depression and despite the decades of research on antidepressants, there is still a need for new drugs with higher efficacy, a faster onset of action, and a better safety profile [20,21]. Access to effective treatments would improve the patients’ quality of life and reduce the economic burden caused by depression-related disability and incapacity to work.

In recent years, psychopharmacologists and psychiatrists working in the field of depression have become interested in drugs currently used for other indications. The most widely studied and most promising drug in this group is ketamine, known mainly as an anesthetic with analgesic and anxiolytic properties [22,23]. The cholinergic hypothesis of depression proposed in the early 1970s and the observations that the acetylcholinesterase inhibitor physostigmine increases depressive mood in patients with unipolar and bipolar depression [24,25], have stirred interest in nonselective muscarinic antagonists such as scopolamine [26,27].

Scopolamine is an anticholinergic and antispasmodic drug approved in more than 80 countries for the prevention of nausea and vomiting caused by motion sickness and for the prevention of postoperative nausea and vomiting. It is available in oral and intravenous (IV) forms or as a transdermal patch [28,29,30]. In terms of the mechanism of action, scopolamine is a non-specific antagonist of muscarinic acetylcholine receptors (mAChRs), which belong to the class A (also known as rhodopsin-like) G protein-coupled receptors superfamily. There are five mACh receptors, each demonstrating a distinct distribution pattern and G-protein coupling/signaling profile. The muscarinic M1, M3, and M5 acetylcholine receptors are generally postsynaptic and excitatory by nature, signaling primarily through the Gαq/11 subset of G proteins. On the other hand, the presynaptic M2 and M4 autoreceptors couple to the Gαi/o subset of G proteins and are inhibitory [31,32]. Consequently, when a single ligand such as scopolamine binds to multiple receptor subtypes, it can modulate a broader range of signaling cascades, leading to various physiological responses.

Studies in rodents illustrate that the antidepressant-like behavioral effects of scopolamine require the stimulation of mTORC1 signaling in the prefrontal cortex (PFC) [33]. The key event that preludes mTORC1 activation appears to be a glutamate burst in the PFC [33], caused by the disinhibition of glutamatergic neurons. Subsequently, increased glutamate transmission induced by scopolamine triggers the activation of AMPA receptors [33,34]. This activation leads to an increased level of BDNF [35], which, in turn, stimulates mTORC1 signaling and consequently drives synaptogenesis processes [33,36] by binding to TrkB receptors.

The first small studies on the use of scopolamine in the treatment of patients with depression were published in the late 1980s [37,38]. A systematic review conducted by Jaffe et al. [38] in 2013 revealed the promising results of a few randomized controlled trials (RCTs), in which IV scopolamine was administered at a dose of 4 μg/kg every 3 to 5 days in adult patients, mainly with MDD or bipolar II disorder. Importantly, scopolamine was shown to have a rapid onset of action—within 3 days after the initial infusion. However, so far, scopolamine has not been approved for use in the treatment of depression.

Considering the promising findings reported by Jaffe et al. [38], the aim of our study was to review and summarize the latest scientific evidence from RCTs on the use of scopolamine in the treatment of depression.

## 2. Methods

**Search strategy and selection criteria:** The systematic review was conducted and reported according to the PRISMA Statement for reporting systematic reviews [39]. The search strategy was designed using Medical Subject Heading terms and Emtree combined with Boole logical operators in Medline (via PubMed), Embase, and the Cochrane Central Register of Controlled Trials (CENTRAL) until 5 June 2023. The following combination of keywords was used: (depression OR depressive symptoms OR depressive symptom OR symptom, depressive OR emotional depression OR depression, emotional) AND (scopolamine OR scopolamine hydrobromide OR Hyoscine OR Transderms Scop OR Scopoderm TTS OR Transderm-V OR Transderm V OR Travacalm HO OR Vorigeno OR Boro-Scopol OR Boro Scopol OR Isopto Hyoscine OR Kwells OR Scoburen OR Scopace OR Scopolamine Cooper).

The search was conducted independently by two reviewers (P.M. and K.Ś.) using the same search strategy and selection of studies on the basis of the previously established inclusion criteria. The study selection was based on the title and abstract and, if necessary, on full-text articles. Any discrepancies were resolved by consensus by the third reviewer (P.K.). There was a high degree of agreement between the reviewers (94%).

Studies were selected for inclusion based on the following criteria: (1) English-language RCTs comparing scopolamine with placebo or other active treatment (head-to-head trials); (2) patients with MDD; and (3) studies published since 2012. Nonrandomized or uncontrolled open-label studies, unpublished studies, and conference abstracts were not included because of their lower credibility and the lack of appropriate data or detailed information about the methodology and study results. Full-text articles were included if they contained information about the study population and treatment regimen as well as necessary data to extract.

**Quality appraisal**: Risk of bias of identified RCTs was assessed according to the Risk of Bias Tool 2.0 (RoB2) [40,41]. RoB2 allows an evaluation of the following domains: randomization process, deviations from intended intervention, missing data outcome, measurement of the outcome, and selection of the reported results. The domain-based evaluation allows the assignment of the following ratings to each domain: Low risk of bias (“+”), high risk of bias (“–”), or unclear risk of bias (“?”). The robvis tool was used to graphically present the results of the risk-of-bias assessment for individual trials [42]. The rating scale was conducted by the authors of this review (K.Ś. and P.M.). Two authors systematically assessed each domain and independently estimated the potential risk of bias for each study. Any discrepancies were resolved by consensus by the third reviewer (P.K.).

**Quantitative assessment**: due to the small number of trials and the high heterogeneity among studies, it was not possible to conduct a quantitative analysis (meta-analysis).

## 3. Results

Four studies conducted in adult patients with MDD met the inclusion criteria (Khajavi et al. [43], Park et al. [44], Zhou et al. [45], and Chen et al. [46]) (Figure 1). The characteristics of the studies are presented in Table 1. The quality of the included RCTs was assessed based on the RoB2 tool (Figure 2). The risk of bias was assessed as low for the trial by Khajavi et al. [43] and Chen et al. [46], while some concerns were found for the trials by Park et al. [44] and Zhou et al. [45], mainly because of the lack of information on allocation sequence concealment (Figure 2).

The study by Khajavi et al. [43], conducted in two centers in Iran, included 40 patients with MDD who were divided into two groups of 20 participants each. In the study group, oral scopolamine at a dose of 0.5 mg twice daily was used in combination with oral citalopram at a dose of 20 mg/day for the first week and then at a dose of 40 mg/day. The other group received a placebo (with the same appearance as scopolamine) plus citalopram. Patients were followed for 6 weeks. The primary endpoint was the severity of depression measured by the 17-item Hamilton Depression Rating Scale (HDRS17) as a change from baseline to week 6. The scopolamine group (change from baseline to week 6: −17.9 [95% confidence interval (CI): −19.3; −16.6]) showed a significantly greater reduction in the mean HDRS17 score compared with the placebo group (change from baseline to week 6: −14.7 [95% CI: −16.0; −13.4]). All patients in the scopolamine group and most patients in the placebo group reached a 50% reduction in the HDRS17 score by week 6. However, the remission rate, defined as an HDRS17 score of 7 or lower, was significantly higher in the scopolamine group than in the placebo group. In terms of assessing the safety profile, no attrition or serious adverse events (SAEs) were reported during the study. The most common adverse events (AEs) included dry mouth (scopolamine vs. placebo: 50% vs. 20%), dizziness (40% vs. 15%), and blurred vision (40% vs. 15%). The results are presented in Table 2.

Park et al. [44] conducted a single-center crossover randomized trial with a single-blind placebo lead-in in which 23 patients with MDD were randomized to receive two counterbalanced blocks of three IV infusions of scopolamine (4 μg/kg) and placebo every 3 to 4 days. Of the 23 participants, 11 were assigned to receive placebo and then scopolamine, and 12 were assigned to receive scopolamine and then placebo. The primary endpoint measure was the 10-item Montgomery-Åsberg Depression Rating Scale (MADRS) score, and the secondary endpoint measure was the Hamilton Anxiety Rating Scale score. Primary and secondary outcome results showed no significant differences between scopolamine and placebo in terms of the antidepressant or anxiolytic effects. Response defined as a ≥50% improvement in the MADRS score from baseline was observed in two participants (one per group) during the scopolamine condition. One of these responders also met the criteria for remission (MADRS score ≤ 10). An unclear antidepressant effect of scopolamine was also observed in a participant who met the criteria for response during the scopolamine and placebo administrations. Two participants who did not respond during the scopolamine treatment achieved remission after switching to the placebo. Most patients reported transient minor AEs at the time of infusion (both scopolamine and placebo). The most common adverse events included dry mouth, constipation, blurred vision, drowsiness, and nervousness. No unexpected AEs or SAEs were reported (Table 2).

Zhou et al. [45] conducted a three-arm, double-blind RCT including 66 participants with MDD (22 participants per arm) followed for 4 weeks. While the previous studies included in the systematic review by Jaffe et al. [2] focused mainly on IV scopolamine, Zhou et al. [45] were the first to investigate intramuscular (IM) scopolamine and scopolamine augmentation as an add-on therapy to currently available antidepressants (oral escitalopram, 10 mg/day). Participants were assigned to low-dose scopolamine (0.3 mg IM once daily) + oral escitalopram (10 mg/day) + IM saline once daily, or to high-dose scopolamine (0.3 mg IM twice daily) + oral escitalopram (10 mg/day), or to placebo (oral escitalopram [10 mg/day] + IM saline twice daily). During the 4-week follow-up, 11 participants withdrew from the study. The primary endpoint was the time to early improvement, defined as an at least 20% reduction in the HDRS17 score (change from baseline to week 4). The study showed no differences between the three groups in the average time until designated improvement. The time to early improvement was 3 days (95% CI: 2; 7) in the low-dose group, 3 days (95% CI: 2; 7) in the high-dose group, and also 3 days (95% CI: 3; 4) in the placebo group. The secondary endpoints included response rates (a 50% decrease in the HDRS17 score from baseline), remission rates (HDRS17 score ≤ 7), and changes in HDRS17, MADRS, Quick Inventory of Depressive Symptomatology Self Report 16-item (QIDS-SR16), Generalized Anxiety Disorder 7-Item (GAD-7), and Clinical Global Impression of Severity Scale (CGI-S) scores throughout the trial. The cumulative response and remission rates were 72.7% (48 of the 66 patients) and 47.0% (31 of the 66 patients), respectively. There were no significant differences between groups in the change from baseline to end visit for the total HDRS17, MARDS, QIDS-SR16, GAD-7, and CGI-S scores. Any AEs were reported in 100% of participants in the low-dose group, in 90.9% of participants in the high-dose group, and in 70% of participants in the placebo group. There was a significant difference in the proportion of event type and the total number of AEs between the three groups (*p* = 0.0024). Two participants in the high-dose group dropped out owing to drug-related AEs. Intramuscular scopolamine resulted in a higher incidence of blurred vision, dizziness, drowsiness, and dry mouth. Other AEs included somnolence, nausea, fatigue, anxiety, insomnia, and tachycardia. However, there were no significant differences in terms of AEs between the three groups after the first IM scopolamine injection (Table 2).

The most recent RCT by Chen et al. [46] involved 40 participants with MDD. To our knowledge, this was the first head-to-head study to compare scopolamine and glycopyrronium bromide (active placebo). Participants were assigned to 4 groups with a single IV infusion of scopolamine at a dose of 4 μg/kg, 5 μg/kg, or 6 μg/kg, or glycopyrronium at a dose of 4 μg/kg. The follow-up lasted 6 weeks. The study reported combined results for all patients taking scopolamine. The primary outcome measure was the total MADRS score, and the secondary outcome measure was the total QIDS-SR16 score. The results indicated that scopolamine has no meaningful antidepressant effect in patients with MDD because all groups showed a similar reduction in the MADRS score (by 12.6 ± 8.7 points for scopolamine and by 11.2 ± 9.6 points for glycopyrronium at day 3). On enrollment in the study, most patients had moderate depression. At day 3, the severity of depression changed from moderate to mild. Compared with the glycopyrronium group, the scopolamine group showed higher responder rates at days 1 or 3, but the difference was not significant. Moreover, in terms of the secondary endpoint, the authors reported a change in depression severity from moderate to mild, with a similar effect in both groups: The mean QIDS-SR16 score was reduced from 12.1 ± 4.1 to 7.5 ± 6.2 in the scopolamine group and from 12.1 ± 4.5 to 8.1 ± 3.8 in the glycopyrronium group. The analysis of the safety profile showed that the most common AEs were agitation, dizziness, dry mouth, insomnia, fatigue, and palpitations. Their frequency was similar in both groups. No treatment-emergent SAEs were noted (Table 2).

## 4. Discussion

The results of the high-quality RCTs described above indicate an unclear effect of scopolamine in the treatment of depression. In their systematic review, Jaffe et al. [38] assessed the use of scopolamine based on the results from seven RCTs. They showed that scopolamine is an effective drug both for unipolar and bipolar depression, producing a rapid antidepressant action within 3 days [38]. In addition, they reported that the most common effective regimen in RCTs was IV scopolamine at a dose of 4 µg/kg administered every 3 to 5 days. Our analysis confirmed these findings only for the study of Khajavi et al. [43] conducted in 2012, in which scopolamine was compared with placebo. Importantly, this RCT was the first to investigate the use of oral scopolamine (in the form of tablets) in patients with MDD. The more recent RCTs by Park et al. [44], Zhou et al. [45], and Chen et al. [46] reported ambiguous results in terms of the antidepressant effect of scopolamine.

The discrepancy in findings may be due to several factors: (1) The use of different comparators; (2) different follow-up duration; (3) different administration routes; (4) different doses of scopolamine; (5) concomitant use of other antidepressants; and (6) differences in the gender ratio. In the identified RCTs, placebo (Khajavi et al. [43], Park et al. [44], and Zhou et al. [45]) or glycopyrronium (Chen et al. [46]) was used as a comparator, while the active treatment was administered via different routes: Oral (one RCT Khajavi et al. [43]), IM (one RCT Zhou et al. [45]), or IV (two RCTs Park et al. [44] and Chen et al. [46]). There were also discrepancies in the scopolamine dose and administration regimen: Twice daily vs. once daily. A significant limitation preventing comparison of treatment effectiveness included concomitant medications such as administration of escitalopram (Khajavi et al. [43]) or citalopram (Zhou et al. [45]) or lack of concomitant medications (Park et al. [44] and Chen et al. [46]). There were also slight differences in the male ratio in identified studies (40%—Khajavi et al. [43], 33%—Park et al. [44], 27%—Zhou et al. [45], and 38%—Chen et al. [46]).

In the study by Park et al. [44], scopolamine was not associated with a significant improvement in depression or anxiety symptoms vs. placebo in patients with MDD. It is possible that participants in this study were more resistant to treatment than those in previous studies on scopolamine assessed by Jaffe et al. [38]. Park et al. [44] also noted that the discrepancy may have been due to the increased severity of depressive symptoms, as reflected by the higher mean MADRS score on enrollment in their study (33 vs. 23 and 30 in the previous studies).

In their study, Zhou et al. [45] showed that IM scopolamine administration did not significantly reduce the effective time or improvement of depressive symptoms in patients with moderate to severe MDD vs. placebo. The discrepancy with the results of the systematic review by Jaffe et al. [38], which confirmed a rapid antidepressant effect of IV scopolamine, may be related to the change in the route of injection (from IV to IM). Further studies are needed to investigate the antidepressant effect of scopolamine depending on the route of administration (IV vs. IM vs. oral).

In the most recent study on scopolamine, Chen et al. [46] compared the use of IV scopolamine with an active comparator, glycopyrronium, and not with placebo as in the previous studies. The study showed no differences in antidepressant activity or response rate between scopolamine and glycopyrronium in patients with MDD. However, it should be noted that glycopyrronium is not approved for use in MDD. The authors indicated that glycopyrrolate is a muscarinic receptor antagonist such as scopolamine, but it cannot cross the blood–brain barrier due to its quaternary ammonium compound structure. Therefore, it is a reasonable control that mimics scopolamine in terms of its adverse effect profile and peripheral pharmacological effects [46]. Taking into account the comparison with active placebo (glycopyrronium), the results of the study by Chen et al. [46] should be interpreted with extreme caution.

Chen et al. [46] reported significant differences in the MADRS score (a change from baseline to day 3). In their study, the improvement in the MADRS score was 12.6. In contrast, in the previous studies (three studies included in the systematic review by Jaffe et al. [38] and the studies by Park et al. [44] and Ellis et al. [47]), the improvement ranged from 4.1 to 9.6 when scopolamine was administered first and from 2.6 to 12.5 when it was administered after the crossover. Chen et al. [46] also referred to the study by Ellis et al. [47], but this study did not meet our inclusion criteria because it assessed a different population (MMD and bipolar disorders) and used a different type of analysis (post hoc). Nevertheless, Ellis et al. [47] showed that scopolamine use is associated with a significant response to placebo in the group of treatment-resistant and treatment-naive patients. Treatment-naive patients had lower MADRS scores than patients resistant to treatment. The authors concluded that scopolamine rapidly reduces symptoms both in treatment-resistant and treatment-naive groups and leads to sustained improvement even in treatment-resistant patients [47].

The RCTs included in our analysis were generally well designed (with a low risk of bias or some concerns about the risk of bias assessed by the RoB tool). However, it should be noted that they were conducted on a relatively small group of patients, considering the prevalence of depression worldwide. No major quality concerns were identified in two studies: Khajavi et al. [43] and Chen et al. [46]. In the case of the study by Park et al. [44], there were concerns about quality in terms of the first domain (bias arising from the randomization process—lack of data on allocation) and the last domain (bias in the selection of the reported results). In the study by Zhou et al. [45], there were some concerns due to the lack of the allocation code and the lack of data on the randomization method; overall six of the 66 participants (9.1%) were lost to follow-up.

The strength of our study is its rigorous and conservative methodology, including a clear search strategy and predefined criteria for the inclusion of studies in a systematic review. Only RCTs of the highest quality were considered for analysis. However, our study also has several limitations, including a small number of RCTs and a relatively short follow-up duration (up to 6 weeks). Such a short follow-up may be insufficient to evaluate the long-term effects of scopolamine. Another limitation is the lack of sufficient data from studies directly comparing current antidepressant regimens with scopolamine in patients with depression. Considering these limitations and the differences between the compared studies, our results should be interpreted with caution.

Research into the antidepressant effects of scopolamine is ongoing. Several studies on scopolamine use are currently underway, including the single-site, randomized, double-blind, placebo-controlled, parallel, phase IIb SCOPE-BD study [48] (registered study protocol), which compares IV scopolamine with placebo in patients with bipolar disorder experiencing a depressive episode. Participants are currently being recruited, and the estimated study completion date is scheduled for December 2024. Another double-blind, randomized, controlled parallel design phase IV study (NCT03386448) [49], which assessed the safety and efficacy of scopolamine and naltrexone in the treatment of major depression, showed a significant improvement in the mean MADRS score after 4 weeks of scopolamine and naltrexone administration vs. placebo (12.5 [95% CI: 6; 23] vs. 3.5 [1; 7]; differences between groups: 9.0, *p* = 0.03). However, there is a high level of uncertainty about the results because the study has not been published yet and the size of the study population is very small (14 participants; two participants in the scopolamine and naltrexone groups were withdrawn from the study, resulting in a final sample of 12 participants, six participants per group).

Future Directions: Considering that the available scientific evidence is inconclusive, further research on the antidepressant effects of scopolamine is needed. In particular, RCTs with an active comparator, a larger population of patients, and long-term follow-up should be conducted to provide more robust data.

## 5. Conclusions

The analysis of the available scientific evidence revealed that the antidepressant effect of scopolamine is unclear and even indicated its absence. Previous studies reported antidepressant activity for IV scopolamine vs. placebo. The effects of IM and IV scopolamine administration are ambiguous, and the evidence is available only for a comparison with placebo or active placebo. Oral scopolamine may be associated with beneficial clinical effects; however, it was presented only in one study. There is no evidence comparing scopolamine with current treatment options; therefore, it is difficult to draw conclusions about the benefits or limitations of scopolamine use.

## Figures and Tables

**Figure 1 biomedicines-11-02636-f001:**
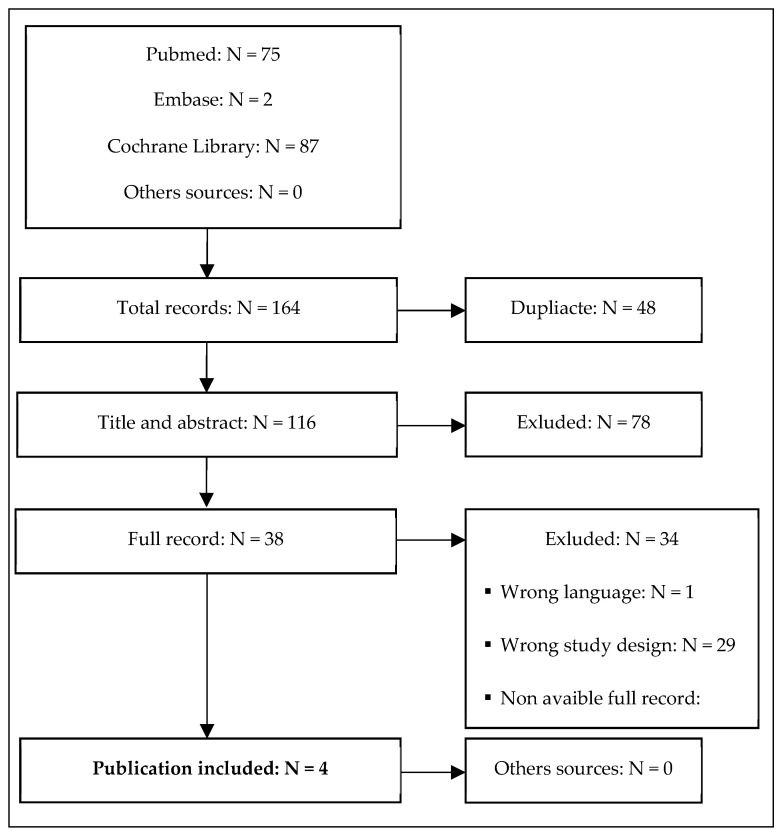
Study flow diagram showing the results of the systematic review and the process of study screening and selection.

**Figure 2 biomedicines-11-02636-f002:**
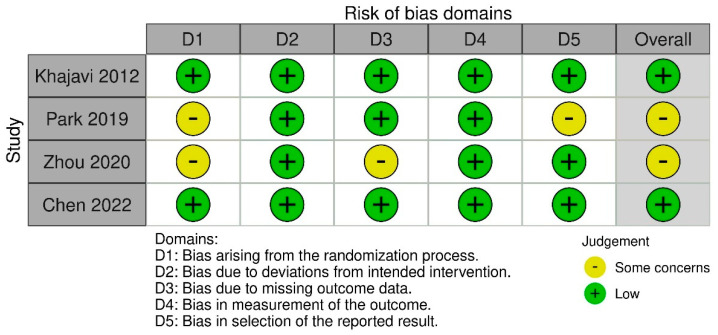
Summary of risk-of-bias assessments for included RCTs. The symbol ‘‘?’’ represents a low risk of bias, ‘‘−’’represents a high risk of bias while the symbol ‘‘?’’ represents an unclear risk of bias [43,44,45,46].

**Table 1 biomedicines-11-02636-t001:** Characteristics of included RCTs regarding the use of scopolamine in the treatment of depression.

Reference	Type of Study	N (Scopolamine vs. Control Group); % Male; Average Age [Years]; Baseline HDRS/HAMA/MADRS Score [Mean ± SD]	Scopolamine vs. Control Group
Khajavi et al. [43]	RCT, two-center, placebo control, double-blind, parallel-group, phase II–III	N = 40 (20 vs. 20) Male: 40% vs. 35% Average age: 37.8 vs. 36.6 Baseline HDRS: 24.5 ± 2.2 vs. 24.2 ± 2.3	Oral scopolamine 0.5 mg twice daily + citalopram 20 mg daily for first week than 40 mg daily for 5 weeks vs. placebo + citalopram 20 mg daily for first week than 40 mg daily for 5 weeks
Park et al. [44]	RCT, single-center, placebo control, single-blind, crossover trial	N = 23 (12 vs. 11) Male: 33% vs. 63% Average age: 40.42 vs. 32.91 Baseline HAM-A: 25.73 ± 8.33 vs. 19 ± 5.67 Baseline MADRS: 34.08 ± 4.25 vs. 31.64 ± 4.2	Scopolamine 4 μg/kg IV /placebo vs. placebo/scopolamine 4 μg/kg IV
Zhou et al. [45]	RCT, single-center, double-blind, parallel-group, three-arm study	N = 66 (22 vs. 22 vs. 22) Male: 27% vs. 32% vs. 45% Average age: 25.7 vs. 26.5 vs. 27.1 Baseline HDRS17 25.7 ± 4.7 vs. 25.4 ± 4.2 vs. 24.5 ± 5.0 Baseline MADRS: 32.2 ± 5.8 vs. 33.5 ± 6.4 vs. 31.0 ± 7.9	Low-dose (scopolamine 0.3 mg IM once daily + oral escitalopram 10 mg/day + IM saline once daily) vs. high-dose (scopolamine 0.3 mg IM twice daily + oral escitalopram 10 mg/day) vs. placebo (oral escitalopram 10 mg/day + IM saline twice daily)
Chen et al. [46]	RCT, double-blind, parallel-group, phase II–III	N = 40 (24 [all scopolamine groups] vs. 16) Male: 38% vs. 19% Average age: 33.0 vs. 37.8 Baseline MADRS: 28.3 ± 4.3 vs. 27.7 ± 4.4	Scopolamine 4 μg/kg IV vs. scopolamine 5 μg/kg IV vs. scopolamine 6 μg/kg IV vs. glycopyrronium bromide 4 μg/kg IV

HDRS—Hamilton Depression Rating Scale; HAM-A—Hamilton Anxiety Rating Scale; IM—intramuscular; IV—intravascular; MADRS—Montgomery-Asberg Depression Rating Scale; RCT—randomized controlled trial.

**Table 2 biomedicines-11-02636-t002:** Main results of included RCTs regarding the use of scopolamine in the treatment of depression.

Reference	Change in MADRS or HDRS, Mean [95% CI]	Response or Remission Rate	Safety—AEs, SAEs
Khajavi et al. [43]	Scopolamine oral vs. placebo (after 42 days): HDRS: −3.2 [−5.1; −1.4], *p* = 0.001	Scopolamine oral vs. placebo (after 42 days): response (50% reduction in HDRS): RR = 0.495 [0.32; 0.65], *p* = 0.231 remission: RR = 0.34 [95% CI: 0.14; 0.83], *p* = 0.004	Scopolamine oral vs. placebo (after 42 days): AEs: no information provided SAEs: 0% vs. 0%
Park et al. [44]	Scopolamine IV vs. placebo: HAM-A and MADRS: no differences between groups	Scopolamine IV vs. placebo: response (50% reduction in MADRS): 8% vs. 0%; remission (MADRS ≤ 10): 4% vs. 0%	Scopolamine IV vs. placebo: AEs: no information provided SAEs: 0% vs. 0%
Zhou et al. [45]	Scopolamine high dose IM vs. placebo: HDRS17: 0.2 [−1.0; 1.5] MADRS: 0.4 [−1.4; 2.2] Scopolamine low dose IM vs. placebo: HDRS17: 0.4 [−0.9; 1.7] MADRS: 0.8 [−0.9; 2.5]	Scopolamine low dose IM vs. scopolamine high dose IM vs. placebo: response (50% reduction in HDRS17) for all groups (cumulative): 72.7% remission (HDRS17 ≤ 7) for all groups (cumulative): 47.0%	Scopolamine low dose IM vs. scopolamine high dose IM vs. placebo: AEs: 100% vs. 90.9% vs. 70%, *p* = 0.0024 SAEs: no information provided
Chen et al. [46]	Scopolamine IV vs. glycopyrronium (placebo): MADRS: no differences between groups	Scopolamine IV vs. glycopyrronium (placebo): response (50% reduction in MADRS at 1 or 3 day): OR = 1.8 [95% CI: 0.6; 5.5] remission: no information provided	Scopolamine IV vs. glycopyrronium (placebo): AEs: no information provided SAEs: 0% vs. 0%

AEs—adverse events, CI—confidence interval, HDRS—Hamilton Depression Rating Scale; HAM-A—Hamilton Anxiety Rating Scale; IM—intramuscular; IV—intravascular; MADRS—Montgomery-Asberg Depression Rating Scale; OR—odds ratio; RR—relative risk, SAEs—serious adverse events.

## Data Availability

Not applicable.

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
