# Peer review of "The Potential of Scopolamine as an Antidepressant in Major Depressive Disorder: A Systematic Review of Randomized Controlled Trials"

_biomedicines, 2023, doi:10.3390/biomedicines11102636_

Round 1
Reviewer 1 Report
Moćko and colleagues, in the present review article titled ‘Antidepressant effects of scopolamine: a review of current evidence’ focus on the use of scopolamine, a drug commonly used for motion sickness and postoperative nausea, as a potential treatment for Major Depressive Disorder (MDD). The introduction highlights the characteristics and global impact of MDD, emphasizing the need for improved treatment options. It introduces scopolamine's mechanism of action as a non-specific antagonist of muscarinic acetylcholine receptors (mAChRs) and its potential in modulating various signaling cascades associated with depression. The study's methods involve a systematic review of randomized controlled trials (RCTs) conducted to assess scopolamine's efficacy in MDD treatment. The search strategy is outlined, including selection criteria for studies. Four RCTs are included in the review. The results section provides a detailed analysis of each study, comparing scopolamine's effects with placebos or other active treatments. The RCTs present mixed findings regarding scopolamine's antidepressant effects, varying by administration route, dosage, and study design. The discussion section synthesizes the reviewed studies and discusses the ambiguous nature of scopolamine's antidepressant effects. While some studies suggested positive outcomes for scopolamine, others showed no significant improvement in depressive symptoms compared to placebos or active comparators. The authors note potential factors contributing to these discrepancies, including differences in administration routes, doses, and study populations. The conclusion underscores the uncertainty surrounding scopolamine's efficacy as an MDD treatment. The authors emphasize the need for further research, particularly addressing the variability in study outcomes and potential biases. They highlight ongoing studies and trials that are exploring scopolamine's antidepressant effects, acknowledging the complexities in understanding its therapeutic potential.
In general, I think the idea of this article is really interesting and the authors’ fascinating observations on this timely topic may be of interest to the readers of Biomedicines. However, some comments, as well as some crucial evidence that should be included to support the author’s argumentation, needed to be addressed to improve the quality of the manuscript, its adequacy, and its readability prior to the publication in the present form. My overall judgment is to publish this paper after the authors have carefully considered my suggestions below, in particular reshaping parts of the ‘Introduction’ and ‘Methods’ sections by adding more evidence.
Please consider the following comments:
• I recommend revising the title. While the title is informative, it could be enhanced to be more engaging or intriguing. Also, I would suggest to add a contextual clue or brief descriptor could give readers a hint about the type of evidence or studies included in the review (e.g., "Clinical Trials," "Recent Research," "Meta-analysis"). A revised title could be: "Exploring Scopolamine's Antidepressant Potential: A Comprehensive Review of Current Clinical Evidence" [1-3].
• A graphical abstract that will visually summarize the main findings of the manuscript is highly recommended.
• Abstract: According to the Journal’s guidelines, this section should be presented as a short summary of about 200 words maximum that objectively represents the article. It should let readers get the gist or essence of the manuscript quickly, prepare the readers to follow the detailed information, analyses, and arguments in the full paper and, most of all, it should help readers remember key points from your paper. Please, consider rewrite this paragraph following these instructions [4].
• Keywords: Please list ten keywords chosen from Medical Subject Headings (MeSH) and use as many as possible in the title and in the first two sentences of the abstract. I would suggest adding “Clinical Trial” and “Treatment Efficacy” as keywords.
• Introduction: The authors need to reorganize this section with several paragraphs made up of about 1000 words, introducing information on the main constructs of this study, which should be understood by a reader in any discipline, and making it persuasive enough to put forward the main purpose of the current research the authors have conducted and the specific purpose the authors have intended by this protocol. I would like to encourage the authors to present the introduction starting with the general background, proceeding to the specific background on the definition of MDD, its symptoms, prevalence, and significance as a global health issue. Those main structures should be organized in a logical and cohesive manner [5].
• In this regard, I believe that the Introduction section would benefit from additional information to enhance its clarity and contextualization. To strengthen this section, I believe that it is essential to consider the neural substrates implicated in the pathophysiology of MDD. The introduction could be extended to briefly mention that MDD's etiology involves dysregulations in various neural pathways, including those associated with mood regulation, reward processing, and stress response. This could encompass a mention of the monoamine hypothesis involving serotonin, norepinephrine, and dopamine dysregulation. Additionally, the introduction might touch upon emerging research on the involvement of glutamatergic and neuroinflammatory pathways in MDD [6-7]. Furthermore, considering the focus on scopolamine as a potential treatment, acknowledging its possible interactions with specific neural circuits and neurotransmitter systems associated with depression could enhance the paper's relevance. This neural perspective would contextualize scopolamine's mechanism of action and its potential to modulate signaling cascades within the central nervous system. By incorporating a brief discussion of the neural substrates involved in MDD, the introduction can bridge the gap between the clinical and mechanistic aspects of depression and set the stage for a more nuanced exploration of scopolamine's potential as a therapeutic intervention [8-10].
• Methods: I believe that this section would benefit from a clearer structure and better organization of the flow of information. For example, I believe that the section should include more detailed explanations regarding the rationale for inclusion and exclusion criteria. Additionally, while the search strategy is briefly described, it might be helpful to elaborate on how the chosen keywords and search operators were derived and why specific databases were chosen. Finally, I suggest addressing potential publication bias or limitations arising from the inclusion of only English-language studies and the exclusion of non-randomized or open-label trials.
• Results: Please provide more critical analysis and interpretation of the study results. Are there patterns or trends across the studies? Are there any limitations or biases that could affect the results?
• Discussion: In my opinion, here authors should discuss the implications of the mixed findings on the clinical management of MDD. Address the potential impact of scopolamine on current treatment approaches and highlight its potential benefits or limitations.
• In my opinion, the ‘Conclusions’ paragraph would benefit from some thoughtful as well as in-depth considerations by the authors, because as it stands, it lists down all the main findings of the research, without really stressing the theoretical significance of the study. Authors should make an effort, trying to explain the theoretical implication as well as the translational application of their research.
• In according to the previous comment, I would ask the authors to include a proper and defined ‘Limitations and future directions’ section before the end of the manuscript, in which authors can describe in detail and report all the technical issues brought to the surface.
• References: Authors should consider revising the bibliography, as there are several incorrect citations. Indeed, according to the Journal’s guidelines, they should provide the abbreviated journal name in italics, the year of publication in bold, the volume number in italics for all the references.
I hope that, after these careful revisions, the manuscript can meet the Journal’s high standards for publication. I am available for a new round of revision of this article.
I declare no conflict of interest regarding this manuscript.
Best regards,
Reviewer
References:
1. https://plos.org/resource/how-to-write-a-great-title/
2. https://www.nature.com/nature-index/news-blog/how-to-write-a-good-research-science-academic-paper-title
3. https://www.indeed.com/career-advice/career-development/catchy-title
4. https://www.mdpi.com/journal/biomedicines/instructions
5. https://dept.writing.wisc.edu/wac/writing-an-introduction-for-a-scientific-paper/
6. DOI: 10.17219/acem/165944
7. https://doi.org/10.3390/ijms24065926
8. DOI: 10.3390/biomedicines11030945
9. https://doi.org/10.3389/fnmol.2023.1217090
10. https://doi.org/10.3390/biomedicines11051248
Minor editing of English language required.
Author Response
Dear Reviewers
thank you for your comments.
Please see below our answers to the comments provided.
Rev. 1 comments
Moćko and colleagues, in the present review article titled ‘Antidepressant effects of scopolamine: a review of current evidence’ focus on the use of scopolamine, a drug commonly used for motion sickness and postoperative nausea, as a potential treatment for Major Depressive Disorder (MDD). The introduction highlights the characteristics and global impact of MDD, emphasizing the need for improved treatment options. It introduces scopolamine's mechanism of action as a non-specific antagonist of muscarinic acetylcholine receptors (mAChRs) and its potential in modulating various signaling cascades associated with depression. The study's methods involve a systematic review of randomized controlled trials (RCTs) conducted to assess scopolamine's efficacy in MDD treatment. The search strategy is outlined, including selection criteria for studies. Four RCTs are included in the review. The results section provides a detailed analysis of each study, comparing scopolamine's effects with placebos or other active treatments. The RCTs present mixed findings regarding scopolamine's antidepressant effects, varying by administration route, dosage, and study design. The discussion section synthesizes the reviewed studies and discusses the ambiguous nature of scopolamine's antidepressant effects. While some studies suggested positive outcomes for scopolamine, others showed no significant improvement in depressive symptoms compared to placebos or active comparators. The authors note potential factors contributing to these discrepancies, including differences in administration routes, doses, and study populations. The conclusion underscores the uncertainty surrounding scopolamine's efficacy as an MDD treatment. The authors emphasize the need for further research, particularly addressing the variability in study outcomes and potential biases. They highlight ongoing studies and trials that are exploring scopolamine's antidepressant effects, acknowledging the complexities in understanding its therapeutic potential.
In general, I think the idea of this article is really interesting and the authors’ fascinating observations on this timely topic may be of interest to the readers of Biomedicines. However, some comments, as well as some crucial evidence that should be included to support the author’s argumentation, needed to be addressed to improve the quality of the manuscript, its adequacy, and its readability prior to the publication in the present form. My overall judgment is to publish this paper after the authors have carefully considered my suggestions below, in particular reshaping parts of the ‘Introduction’ and ‘Methods’ sections by adding more evidence.
Please consider the following comments:
- I recommend revising the title. While the title is informative, it could be enhanced to be more engaging or intriguing. Also, I would suggest to add a contextual clue or brief descriptor could give readers a hint about the type of evidence or studies included in the review (e.g., "Clinical Trials," "Recent Research," "Meta-analysis"). A revised title could be: "Exploring Scopolamine's Antidepressant Potential: A Comprehensive Review of Current Clinical Evidence" [1-3].
- The title of the article has been clarified according to suggestion.
- A graphical abstract that will visually summarize the main findings of the manuscript is highly recommended.
- Within the framework of this work, there is no justification for making a graphical abstract. A small number of studies and non-comparable results do not allow for a simple presentation of the results of the work in the form of a graphical abstract.
- Abstract: According to the Journal’s guidelines, this section should be presented as a short summary of about 200 words maximum that objectively represents the article. It should let readers get the gist or essence of the manuscript quickly, prepare the readers to follow the detailed information, analyses, and arguments in the full paper and, most of all, it should help readers remember key points from your paper. Please, consider rewrite this paragraph following these instructions [4].
- We have provided changes to the abstract as suggested - e.g. acronyms have been removed, method aspects have been expanded and the purpose of the study has been added.
- Keywords: Please list ten keywords chosen from Medical Subject Headings (MeSH) and use as many as possible in the title and in the first two sentences of the abstract. I would suggest adding “Clinical Trial” and “Treatment Efficacy” as keywords.
- We added more keywords, and changed the entries in the abstract
- Introduction: The authors need to reorganize this section with several paragraphs made up of about 1000 words, introducing information on the main constructs of this study, which should be understood by a reader in any discipline, and making it persuasive enough to put forward the main purpose of the current research the authors have conducted and the specific purpose the authors have intended by this protocol. I would like to encourage the authors to present the introduction starting with the general background, proceeding to the specific background on the definition of MDD, its symptoms, prevalence, and significance as a global health issue. Those main structures should be organized in a logical and cohesive manner [5]. In this regard, I believe that the Introduction section would benefit from additional information to enhance its clarity and contextualization. To strengthen this section, I believe that it is essential to consider the neural substrates implicated in the pathophysiology of MDD. The introduction could be extended to briefly mention that MDD's etiology involves dysregulations in various neural pathways, including those associated with mood regulation, reward processing, and stress response. This could encompass a mention of the monoamine hypothesis involving serotonin, norepinephrine, and dopamine dysregulation. Additionally, the introduction might touch upon emerging research on the involvement of glutamatergic and neuroinflammatory pathways in MDD [6-7]. Furthermore, considering the focus on scopolamine as a potential treatment, acknowledging its possible interactions with specific neural circuits and neurotransmitter systems associated with depression could enhance the paper's relevance. This neural perspective would contextualize scopolamine's mechanism of action and its potential to modulate signaling cascades within the central nervous system. By incorporating a brief discussion of the neural substrates involved in MDD, the introduction can bridge the gap between the clinical and mechanistic aspects of depression and set the stage for a more nuanced exploration of scopolamine's potential as a therapeutic intervention [8-10].
- The Introduction section was reorganized according to suggestions.
- Methods: I believe that this section would benefit from a clearer structure and better organization of the flow of information. For example, I believe that the section should include more detailed explanations regarding the rationale for inclusion and exclusion criteria. Additionally, while the search strategy is briefly described, it might be helpful to elaborate on how the chosen keywords and search operators were derived and why specific databases were chosen. Finally, I suggest addressing potential publication bias or limitations arising from the inclusion of only English-language studies and the exclusion of non-randomized or open-label trials.
- The methods section has been rephrased. It was divided into sections: Search strategy and selection criteria, Quality appraisal, Quantitative assessment. The last two sections have been added with new data. The Methods were extended with an analysis of the quality of evidence, which was performed, however, in the original version it was not presented in the work. We also added PRISMA diagram.
- Results: Please provide more critical analysis and interpretation of the study results. Are there patterns or trends across the studies? Are there any limitations or biases that could affect the results?
- Evidence quality analysis has been added in the results section.
- Discussion: In my opinion, here authors should discuss the implications of the mixed findings on the clinical management of MDD. Address the potential impact of scopolamine on current treatment approaches and highlight its potential benefits or limitations.
- We added new data related to the quality of included RCTs and information about key quality limitations. Included studies do not compare scopolamine with the current treatment approaches so it is not possible to show benefits/limitations of this intervention. In the conclusion section of the paper, we added information about this problem
- In my opinion, the ‘Conclusions’ paragraph would benefit from some thoughtful as well as in-depth considerations by the authors, because as it stands, it lists down all the main findings of the research, without really stressing the theoretical significance of the study. Authors should make an effort, trying to explain the theoretical implication as well as the translational application of their research.
- We performed some changes in the conclusion section, we also added information indicated above
- In according to the previous comment, I would ask the authors to include a proper and defined ‘Limitations and future directions’ section before the end of the manuscript, in which authors can describe in detail and report all the technical issues brought to the surface.
- The limitations of the study were presented in the discussion section. We also added new data related to the quality of included RCTs and information about key quality limitations.
- References: Authors should consider revising the bibliography, as there are several incorrect citations. Indeed, according to the Journal’s guidelines, they should provide the abbreviated journal name in italics, the year of publication in bold, the volume number in italics for all the references.
- The references were modified in accordance with the requirements of the Journal’s guidelines and also some new references were added.

Reviewer 2 Report
Comments and Suggestions for Authors
Title: “Antidepressant effects of scopolamine: a review of current evidence”.
Paweł Moćko, et al in this review article has described the antidepressant effects of scopolamine using I.V, I.M and oral scopolamine in combination with other antidepressants drugs in randomized controlled trials (RCTs). The review is well designed and written in scientific way. The author should address all the concerns before publishing.
1. Line 33; the references 1 and 2 should be properly cited according to the journal reference format.
2. Para one of introduction should be elaborated more providing updated relevant references, for guidance, https://doi.org/10.3390/ biomedicines10102385, https://doi.org/ 10.3390/biomedicines10102597, doi: 10.7860/JCDR/2014/10258.5292.
3. Provide appropriate references at the end of line 36.
4. Line 52; remove space after the word “symptoms”.
5. Line 57; references should be cited as per the journal standard format [1, 2, 10, 11] and should be apply for all references in this review.
6. This review is limited to the antidepressant effects produced by scopolamine in combination with other antidepressant drugs, then how we conclude from the results of this review that how much the scopolamine will be effective alone as antidepressant.
7. The author should expand the results by introducing updated research studies using scopolamine alone as antidepressant drug.
8. This review lack any study indicating the use of scopolamine in animals models. The author should consider this point as in most of the available literature showed the stress producing effects of scopolamine in a number of animal studies.
9. Line 285; the word “antidepressant” is written in bold font, make it unbold.
10. Line 297, 298, 299, 300; font size was use larger than normal “(12.5 [95% CI: 6; 23] vs 3.5 [1; 7]……………………”, make it adjust accordingly.
11. Use of oral scopolamine as antidepressant was reported in only one study which is not rational.
12. The author should provide the limitation of this study.
13. The author should include more relevant RCTs studies using scopolamine as antidepressant.
14. In my opinion this review is deficient of the relevant data regarding antidepressant effects of scopolamine.
15. The references did not cite according to the journal format, it should be revised accordingly in both in the text and in bibliography.
16. This review has some beneficial aspects for future perspectives after addressing all the concerns.
Moderate editing of English language required
Author Response
Rev. 2 comments
Depression is a global health problem that affects millions of people worldwide. Despite the availability of various treatment options, depression remains a major challenge in the field of mental health. Therefore, there is a need for novel therapeutic agents that can provide faster and more effective relief for patients with depression. In this manuscript, entitled “Antidepressant effects of scopolamine: a review of current evidence,” Moćko and colleagues explore the potential of scopolamine, a drug traditionally used to treat motion sickness, as a promising new treatment option for depression.
This manuscript's main advantage is that it presents a comprehensive review of the latest scientific evidence on the use of scopolamine as a potential therapeutic agent for depression. The authors have conducted a thorough analysis of randomized controlled trials, highlighting the promising antidepressant effects of scopolamine due to its anticholinergic activity. The manuscript also includes a detailed discussion of the potential mechanisms of action of scopolamine and its advantages over traditional antidepressants.
In general, I think the idea of this research article is interesting, and the authors’ fascinating observations on this timely topic may be of interest to the readership of Biomedicines. However, some comments as well as some crucial evidence should be included to support the author’s argumentation and improve its adequacy, readability, and thus the quality of the manuscript prior to publication. My overall opinion is to publish this research article after the authors have carefully considered the reviewers’ comments and my suggestions below during the peer-review session.
Comments:
- First, I would like the authors to clarify the type of review article, such as systematic, scoping, synthetic, rapid, state-of-the art, or narrative, and declare it in the abstract and the objectives. Then, I would like the authors to make sure there are all elements necessary for a certain type of review paper by using the checklist [1–3].
- The article is a review (summary of knowledge) based on a systematic review of the literature – a comprehensive review of current clinical evidence.
- Title: This is the most important section of the manuscript. Please present a concise and self-explanatory title stating the most important findings of this study.
- The title of the article has been clarified.
- Keywords: Please list six keywords from Medical Subject Headings (MeSH) and use as many as possible in the title and in the first two sentences of the abstract.
- We added more keywords, and changed the entries in the abstract
- Abstract: In my opinion, the authors should consider rephrasing this section. According to the Journal’s guidelines, the abstract should contain most of the following kinds of information in brief form: Please consider giving a more synthetic overview of the paper's key points: I would suggest rephrasing the results and conclusion to make them easier for readers to understand. Also, in my opinion, abbreviations should not be used in this section. That having said, I would like the authors to make as much effort for this section as for the rest of the manuscript. Please present the abstract in 200 words (preferably 200–220 words, max. 250) according to the guidelines of the journal, focusing on proportionally presenting the background, methods, results, and conclusion (without the headings of subsections) if it is a systematic review, including a state-of-the-art, rapid, scoping, and synthetic review. If it is a narrative review, the abstract should consist of the introduction, a short summary, and conclusion. The background should include the general background (one to two sentences), the specific background (two to three sentences), and the current issue addressed to this study (one sentence), leading to the objectives. In this subsection, I would like the authors to lay out basic information, a problem statement, and their motivation to break off. The methods should clarify the authors’ approach, such as study design and variables, to solve the problem and/or make progress on the problem. The results should close with a single sentence putting the results in more general context. The conclusion should open with one sentence describing the main result using such words like “Here we show”, which should be followed by statements such as the potential and the advance this study has provided in the field and finally a broader perspective (two to three sentences) readily comprehensible to a scientist in any discipline.
- We have made changes to the abstract as suggested - e.g. acronyms have been removed, method aspects have been expanded and the purpose of the study has been added.
- A graphical abstract is highly recommended.
- Within the framework of this work, there is no justification for making a graphical abstract. A small number of studies and non-comparable results do not allow for a simple presentation of the results of the work in the form of a graphical abstract.
- Introduction: The authors need to fully reorganize this section with several paragraphs made up of about 1000 words, introducing information on the main constructs of this protocol, which should be understood to a reader in any discipline and make persuasive enough to put forward the main purpose of current research the author has conducted and the specific purpose the author has intended by this protocol. I would like to encourage the authors to present the introduction starting with the general background, proceeding to the specific background, rationales, and finally the current issue addressed to this study, leading to the objectives. Those main structures should be organized in a logical and cohesive manner.
- The Introduction section was reorganized according to suggestions.
- Discussion: I would like the authors to present this section with several paragraphs containing approximately 1,500 words, beginning with an introductory paragraph and then providing a summary of the previous section (Results). Then, I anticipate the authors to develop arguments clarifying the potential of this study as an extension of previous work, the implication of the findings, how this study could facilitate future research, the ultimate goal, the challenge, the knowledge and technology required to achieve this goal, the statement regarding this field in general, and the significance of this line of research. It is essential to describe the study's limitations, merits, and potential clinical applications.
- We added new data related to the quality of included RCTs and information about key quality limitations. Included studies do not compare scopolamine with the current treatment approaches so it is not possible to show benefits/limitations of this intervention. In the conclusion section of the paper, we added information about this problem
- Conclusion: I believe that this manuscript would benefit from a single paragraph of 150–200 words that presents some thoughtful and in-depth considerations by the authors as experts in order to convey the main message. The authors should make an effort to explain the theoretical implications as well as the translational application of their research. In order to understand the significance of this field, I believe it would be necessary to discuss theoretical and methodological avenues in need of refinement as well as future research directions.
- We made some changes in the conclusion section, we also added information indicated above
- References: Please follow the guidelines of the journal [4]. Please cite more references, Typically, a review article like this cites over 150 references.
- We added some new references to the paper, however, due to the purpose of the study and the number of identified randomized trials, a significant increase in the number of references was impossible.
Overall, the manuscript has no figures, two tables, and 29 references. I believe that the manuscript may have merit in presenting a well-written and well-researched article that provides valuable insights into the potential use of scopolamine as a novel therapeutic agent for depression. The authors have thoroughly reviewed the most recent scientific research on this subject, and a long list of references backs up their analysis. The manuscript is well-structured, with clear headings and subheadings that make it easy to navigate. The authors have also provided a detailed discussion of the potential mechanisms of action of scopolamine and its advantages over traditional antidepressants. I hope that, after careful revisions, the manuscript can meet the journal’s high standards for publication.
- Thank you for your positive feedback.
Round 2
Reviewer 1 Report
Dear Authors,
Thank you for your thoughtful responses to the comments I provided on the revised manuscript. I appreciate your perspective on the points of disagreement.
After careful consideration, I am satisfied with the revisions you have made and your responses to my concerns. Therefore, I am confirming my acceptance of the manuscript for publication. I believe your efforts have improved the quality of the paper, and I look forward to seeing it contribute to the scientific community.
Best regards,
Reviewer
Author Response
Rev 1
Thank you for your thoughtful responses to the comments I provided on the revised manuscript. I appreciate your perspective on the points of disagreement.
After careful consideration, I am satisfied with the revisions you have made and your responses to my concerns. Therefore, I am confirming my acceptance of the manuscript for publication. I believe your efforts have improved the quality of the paper, and I look forward to seeing it contribute to the scientific community.
Thanks to Rev1 for the manuscript final approval
Reviewer 2 Report
Suggestions incorporated.
Author Response
Rev 2
Incorporated (in text)
All our comments and changes were implemented to the text of the manuscript.